# An Investigation of the Protein Quality and Temporal Pattern of Peripheral Blood Aminoacidemia following Ingestion of 0.33 g·kg^−1^ Body Mass Protein Isolates of Whey, Pea, and Fava Bean in Healthy, Young Adult Men

**DOI:** 10.3390/nu15194211

**Published:** 2023-09-29

**Authors:** Marta Kozior, Robert W. Davies, Miryam Amigo-Benavent, Ciaran Fealy, Philip M. Jakeman

**Affiliations:** 1Department of Physical Education & Sports Sciences, University of Limerick, Limerick V94 T9PX, Ireland; 2Health Research Institute, University of Limerick, Limerick V94 T9PX, Ireland; r.davies@chester.ac.uk (R.W.D.); miryam.amigobenavent@ul.ie (M.A.-B.); 3Chester Medical School, University of Chester, Shrewsbury CH1 4BJ, UK; 4Department of Biological Sciences, University of Limerick, Limerick V94 T9PX, Ireland

**Keywords:** sustainable diet, plant protein, protein quality, aminoacidemia, postprandial, young adults

## Abstract

An increase in the intake of legumes is recommended in the promotion of plant-sourced (PSP) rather than animal-sourced (ASP) protein intake to produce a more sustainable diet. This study evaluated the quality of novel PSP isolates from pea (PEA) and fava bean (FAVA) and an ASP isolate of whey (WHEY) and compared the magnitude and temporal pattern of peripheral arterial aminoacidemia following ingestion of 0.33 g·kg^−1^ body mass of protein isolate in healthy young adult men (n = 9). Total indispensable amino acids (IAA) comprised 58% (WHEY), 46% (PEA), and 42% (FAVA) of the total amino acid (AA) composition, with the ingested protein providing 108% (WHEY), 77% (PEA), and 67% (FAVA) of the recommended *per diem* requirement of IAA. Reflecting the AA composition, the area under the curve (∆AUC_0-180_), post-ingestion increase in total IAA for WHEY was 41% (*p* < 0.001) and 57% (*p* < 0.001) greater than PEA and FAVA, respectively, with PEA exceeding FAVA by 28% (*p* = 0.003). As a sole-source, single-dose meal-size serving, the lower total IAA for PEA and FAVA would likely evoke a reduced post-prandial anabolic capacity compared to WHEY. Incorporated into a food matrix, the promotion of PSP isolates contributes to a more sustainable diet.

## 1. Introduction

Among Irish adults, as in other countries, approximately 60% of dietary protein intake is from animal-sourced protein (ASP) [1,2]. However, a greater plant-sourced protein (PSP) intake is promoted to produce a more sustainable diet [3] and to improve human and planetary health [4,5,6]. Current global PSP intake data [7] indicate that cereal crops (e.g., wheat, rice, and maize) are generally overconsumed, whereas legumes are under-consumed at an unbalanced ratio of 6:1 (i.e., 6 g of cereal protein is consumed per 1 g of legume protein). In addition, legumes contain 20–30% of protein by dry weight [8]. An increase in the intake of legumes to 50 g of pulses and 25 g of soybeans per day is recommended within an overall target of a threefold increase in current consumption by 2050 [3]. To better inform and support the transition to a higher intake of legume-based proteins, a greater understanding of the nutrient quality and functional properties of legumes and legume-based protein derivatives is required [9].

Dietary protein quality is defined by source, amino acid composition, and specific requirements for nutritionally indispensable amino acids (IAA) and limiting digestible indispensable amino acid score (DIAAS) [10]. As with most PSPs, legumes tend to be deficient in one or more essential amino acids, typically the sulphur-containing amino acids methionine and cysteine, are of low digestibility, and contain anti-nutritional factors (ANFs) (e.g., protease inhibitors, lectins, polyphenols, tannins, and phytic acid) [10] that impact the bioavailability of amino acids and protein quality [11,12]. However, commercial processing techniques (i.e., protein extraction and purification) [13] can reduce or inactivate ANFs to generate protein isolates (~80% *w*/*w* protein) available commercially as a food ingredient and/or nutritional supplement. To date, the protein quality and bioavailability of legume-based protein isolates is not widely reported.

Therefore, this study sought to determine the protein quality and temporal change in aminoacidemia following ingestion of a protein-matched bolus of two novel, single-source protein isolates obtained from pea (PEA) and fava bean (FAVA) in comparison to the milk protein whey (WHEY) in healthy young adult men.

## 2. Materials and Methods

### 2.1. Ethical Approval and Participant Recruitment

The study was granted ethical approval by the University of Limerick Education and Health Sciences Research Ethics Committee (2019_12_02 EHS), conducted in accordance with the ethical standards outlined in the most recent version of the Declaration of Helsinki, and registered with clinicaltrials.gov with the identifier NCT04872530. Potential participants were informed of the risks and benefits before providing written informed consent. Eligibility criteria: (i) male aged 18 to 35 y; (ii) healthy (i.e., no current injury, illness, medication, history of chronic disease, or known allergies and intolerances, normotensive, non-obese, with normal blood chemistry). All participant volunteers were screened using a medical questionnaire, physical examination, dietary intake record, anthropometry, and body composition analysis (Tanita MC, 180-MA, Tanita Ltd., Manchester, M1 2HY, UK). Appendix A presents a flow diagram of participant recruitment, enrolment, completion, and analysis.

### 2.2. Study Design

The study investigated temporal change in circulating amino acids following ingestion of 0.33 g·kg^−1^ body mass of a protein isolate sourced from either PEA (*Pisum sativum,* 100% non-GMO, Canadian), FAVA (*Vicia faba* L., Marigot Ltd. Carrigaline, Co. Cork, Ireland), or WHEY (Carbery Group, Ballineen, Co. Cork, Ireland). The composition of the protein isolates is provided in Appendix A. In a randomised, double-blind design, participants maintained their habitual diet and refrained from purposeful exercise and alcohol consumption for the previous 24 h, on three separate occasions at least three days apart. On trial days, participants arrived at the laboratory following an overnight fast.

Participants remained seated throughout each trial. After insertion of a cannula into a superficial vein on the dorsal surface of the hand, the hand was placed in a heated hand box (air temperature 55 °C) for 15 min to arterialise venous drainage. Following a baseline blood draw, participants ingested one of three protein isolates (0.33 g·kg^−1^ body mass) mixed with water in a 1 to 10 ratio within 5 min. Serial samples of arterialised venous blood were drawn pre-ingestion and 15, 30, 45, 60, 75, 90, 120, 150 and 180 min post-ingestion.

### 2.3. Blood Sample Collection and Analysis

Blood was collected into EDTA S-Monovettes^®^ (Sarstedt, Nümbrecht, Germany), centrifuged at 2465 g at 20 °C for 10 min, and aliquots of plasma stored at −80 °C prior to analysis as reported previously [14]. Briefly, following acid extraction, the concentration of amino acid o-phthalaldehyde derivatives was measured by reversed-phase high-pressure liquid chromatography (Agilent Technologies, Santa Clara, CA, USA) on a 4.6 × 50 mm, 1.8 µm, C_18_ rapid resolution column (ZORBAX, Agilent Technologies Ireland Ltd. Cork, Ireland) thermo-stated at 40 °C. Norvaline was used as an internal standard, and quantification of amino acids was carried out using amino acid standards (Agilent Technologies Inc.). Recovery >85% was achieved for spiked and non-spiked pool plasma with an intra-assay coefficient of variation (CV) of <4%.

### 2.4. Treatment of Data and Statistical Analyses

Following the repeated measures design, data were analysed for differences in magnitude of response following ingestion of three protein isolates of WHEY, PEA, and FAVA protein for total (ΣAA) and indispensable (ΣIAA) amino acids. The maximum concentration (C_max_) and change in maximum concentration from baseline concentration (∆C_max_) were determined for each analyte. The overall response of an analyte with respect to time was calculated as the area under the curve of the change in concentration from baseline to 180 min post-ingestion (∆AUC_0-180_). Prior to statistical analysis, potential outliers were assessed by boxplot, and data were assessed for normal distribution by the Shapiro-Wilk test (*p* > 0.05), homogeneity of variance by Levene’s test (*p* > 0.05), and Mauchly’s test of sphericity (*p* > 0.05). Data are reported as the mean (SD) unless stated otherwise. An *a priori* hypothesis for statistical analysis (H_0_) assumed the mean response for all protein isolates was equal (i.e., µ_WHEY_ = µ_PEA_ = µ_FAVA_) and analysed by repeated measures analysis of variance in SPSS Version 28 (IBM Corporation, Armonk, NY, USA).

## 3. Results

### 3.1. Participants

10 participant volunteers were recruited, of which nine completed all three protein trials.

Participant characteristics are provided in Table 1.

### 3.2. Amino Acid Composition and Limiting Digestible Indispensable Amino Acid Content (DIAA) Score of the Protein Isolates

The composition of the protein isolates is provided in Appendix A. Of note, the protein content in the isolates was measured as %N *w*/*w* by combustion or the Kjeldahl method and reported as %mass (dry basis). The measurement of %N as a proxy for total protein requires that a nitrogen-to-protein conversion factor (NPCF), normally 6.25, is applied to calculate the total protein [15]. To attain 0.33 g protein∙kg^−1^ body mass, the required mean protein intake per trial was 26.4 (2.7) g. Correcting for the protein content of each isolate, the amount of protein powder mixed into 500 mL and fed to participants varied from 34 (3.5), 31.9 (3.3), to 34.4 (3.5) g for FAVA, PEA, and WHEY isolates, respectively.

The contribution of ingested protein to the most recent recommended estimate of adult indispensable amino acid requirements, an indicator of protein quality, is presented in Table 2. The indispensable amino acid composition of the protein isolates under investigation ranged from 52% for WHEY to 46% and 42% for PEA, and FAVA, respectively. Ingestion of 0.33 g protein∙kg^−1^ body mass provided 108% (WHEY), 77% (PEA) and 67% (FAVA) of the recommended *per diem* total IAA intake. No specified requirement for individual dispensable amino acid is currently available to undertake a similar comparison. In addition, Appendix A provides an estimate of the most limiting digestible indispensable amino acid score (DIAAS) for WHEY (129% (Methionine)), PEA (62% (Methionine)), and FAVA (55% (sulphur amino acids, SAA)). DIAAS incorporated published values of the true ileal digestibility (TID) of IAA for whey, pea, and fava bean [16,17,18].

### 3.3. Temporal Pattern of Change in Arterialised Amino Acid following Ingestion of Protein Isolates

There was no difference in the mean pre-ingestion basal arterial [AA] between trials. The temporal appearance, absolute and relative change in total (ΣAA) and indispensable (ΣIAA) amino acids following ingestion of the protein isolates are presented in Figure 1.

The post-ingestion change in total amino acids (ΣAA) followed a similar pattern of response for all protein isolates, with the peak concentration (C_max_) occurring between 30 and 60 min, with an indication of an earlier peak response for PEA. Values for the mean peak concentration (C_max_) and overall response, represented by the post-ingestion integrated area under curve (∆AUC_0-180_), are presented in Table 3. By comparison, the mean C_max_ for ΣAA is higher for WHEY and PEA, with WHEY exceeding PEA by 6% (NS), FAVA by 16% (*p* < 0.001) and PEA exceeding FAVA by 11% (*p* < 0.005). Represented by the ∆AUC_0-180,_ the overall post-ingestion increase in ΣAA for WHEY was 22% (*p* < 0.001), and 37% (*p* < 0.001) greater than PEA and FAVA, respectively, with PEA exceeding FAVA by 19% (*p* = 0.003).

C_max_ of ΣIAA was 33% (*p* < 0.001) higher for WHEY than PEA, 34% (*p* < 0.001) higher for WHEY than FAVA, with PEA 25% (*p* < 0.001) higher than FAVA. Overall, the ∆AUC_0-180_ post-ingestion increase in ΣIAA for WHEY was 41% (*p* < 0.001) and 57% (*p* < 0.001) greater than PEA and FAVA, respectively, with PEA exceeding FAVA by 28% (*p* = 0.003).

## 4. Discussion

Great importance is attached to the amount and quality of protein required to meet human nutritional needs and the protein supplied by food ingredients, whole foods, sole-source foods, and mixed diets to meet those needs [5,6]. This study assessed the protein quality and quantified the magnitude of the temporal change in circulating amino acids following ingestion of an equal amount of commercially available protein isolates of PEA, FAVA, and WHEY in young, healthy adult men. To obtain an equal amount of protein, the amount of powdered isolate was adjusted for the %N *w*/*w* estimate of protein composition. As the %N per amino acid varies from 7.73% for tyrosine to 32.16% for arginine, the %N of the total amino acid content per unit mass varies in proportion to the amino acid composition of each isolate, i.e., 11.35, 10.97, and 9.83% for WHEY, PEA, and FAVA, respectively ( Appendix A).

As recommended by the World Health Organization/Food and Agriculture Organization of the United Nations/United Nations University (WHO/FAO/UNU) [5], the protein quality of the isolates was evaluated based on the amino acid composition, the specific requirement for the nutritionally indispensable amino acids (IAA), and the limiting digestible indispensable amino acid scores (DIAAS). Similar to an earlier study of commercially available protein isolates [19], the amino acid composition of the plant-sourced protein isolates of PEA and FAVA were found to contain lower total IAA compared to the animal-sourced protein isolate WHEY. Indeed, IAAs comprised 52% (WHEY), 46% (PEA), and 42% (FAVA) of the total AA composition of the protein isolates under investigation.

In this study, the IAA composition per 100 g of the protein content of each isolate was 57 g for WHEY, 46 g for FAVA, and 39 g for PEA. Based on a mean total protein requirement of 0.66 g∙kg^−1^∙d^−1^ the current recommended estimate of the requirement for IAA is 0.18 g∙kg^−1^∙d^−1^ (5). Feeding 0.33 g∙kg^−1^∙d^−1^ (i.e., a meal-size serving) of protein isolate provided 108% (WHEY), 77% (PEA), and 67% (FAVA) of the recommended 184 mg∙kg^−1^∙d^−1^ intake of IAA (Table 2). The DIAAS incorporate true ileal digestibility, and thereby the bioavailability of individual amino acids. As the DIAAS is calculated per gram of protein, comparative evaluation of DIAAS derived from different protein sources is conducted on an isonitrogenous basis and may be usefully employed when single-source isolates are used as food ingredients, as supplements to habitual diets, or when complementation is a consideration [6]. Defined by the estimate of the DIAAS, WHEY scored within the ‘high/excellent quality’ protein range (DIAAS = 129, i.e., >100), with methionine identified as the limiting IAA, though the DIAAS for histidine had a similar value. PEA and FAVA DIAAS resided within the ‘no quality claim’ category (DIAAS < 75), with methionine + cysteine identified as the limiting IAAs.

Following digestion, the magnitude of peripheral arterial aminoacidemia reflected the difference in the amino acid composition of the protein isolates. Matched for equivalent protein intake, and in contrast to our hypothesis, significantly greater net aminoacidemia (Σ∆AA; AUC_0-180_) was observed for WHEY than PEA (+22%) and FAVA (+37%). In a similar pattern of response, indispensable aminoacidemia (ΣIAA; AUC_0-180_) for WHEY was 41% and 57% greater than PEA and FAVA, respectively.

In addition to the quantifiable difference in gram amount of amino acid intake per unit protein mass, differences attributed to the rate of digestion, absorption, splanchnic retention, and uptake by peripheral tissues [20] may also contribute to the observed response. Though not readily quantifiable, studies tracing the metabolic fate of intrinsically labelled milk protein for 5 h post-ingestion of a meal-like amount of whey protein indicate ~55% of the protein-derived amino acids appear in the circulation [21]. To the authors’ knowledge, no such similar studies have been reported for pea and fava proteins.

The aminoacidemia following ingestion of protein provides the substrate precursors of de novo protein synthesis and, as in the example of leucine, activators of cellular protein synthesis [22]. Considered as a sole-source protein and based on a single dose of 0.33 g of protein per kilogram of body mass (i.e., a meal-size serving), the lower total IAA and limiting IAAs identified by the DIAAS would likely contribute to a reduced post-prandial anabolic capacity for PEA and FAVA compared to WHEY. However, in a recent post-prandial study, ingestion of an exclusively derived complementary plant-sourced protein blend matched the protein synthetic response to an equivalent amount of milk-based protein [23]. In this respect, further incorporation of PSP isolates contributes to the broader objective of producing a more sustainable diet and improving human and planetary health.

## Figures and Tables

**Figure 1 nutrients-15-04211-f001:**
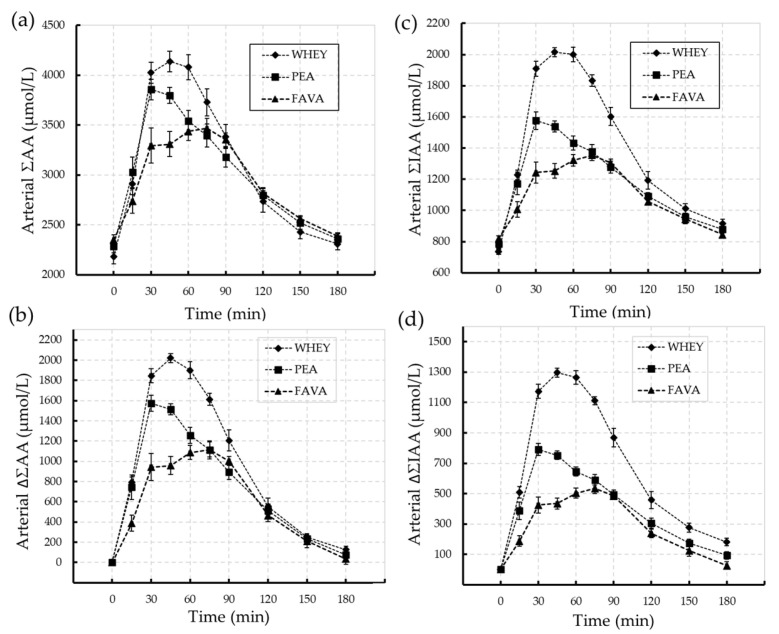
Temporal change in total amino acids (ΣAA; panel (**a**,**b**)) and indispensable amino acids (ΣIAA; panel (**c**,**d**)) following ingestion of the WHEY (
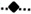
), PEA (
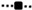
) and FAVA (
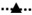
) protein isolates in healthy young adult men (*n* = 9). Data represents the Mean (SEM).

**Table 1 nutrients-15-04211-t001:** Participant characteristics (*n* = 9).

	Mean	SD
Age (y)	24.9	4.1
Height (cm)	179.2	6.2
Body mass (kg)	79.9	8.2
Body mass index (kg.m^−2^)	24.8	1.7
Lean tissue mass (kg)	62.4	6.1
Fat mass (%)	17.7	3.4

**Table 2 nutrients-15-04211-t002:** Relative contribution to estimates of the mean *per diem* amino acid requirements from the ingestion of 0.33 g∙kg^−1^ of protein isolate in young adult men.

	Estimate of Requirement ^1^	WHEY		PEA		FAVA	
	(mg∙kg^−1^∙d^−1^)	(mg∙kg^−1^)	(%)	(mg∙kg^−1^)	(%)	(mg∙kg^−1^)	(%)
Histidine	10	6.9	69	8.8	88	8.1	81
Isoleucine	20	21.6	108	15.2	76	12.8	64
Leucine	39	40.2	103	26.1	67	24.1	62
Lysine	30	34.8	116	24.1	80	20.5	68
Methionine + cysteine	15	16.1	107	10.2	68	5.5	37
*Methionine*	10	6.9	69	3.6	36	2.2	22
*Cysteine*	4	9.2	229	6.6	164	3.3	82
Phenylalanine + tyrosine	25	23.9	96	26.6	107	25.0	100
Threonine	15	27.7	185	11.2	75	10.9	73
Tryptophan	4	5.3	133	2.8	71	2.6	65
Valine	26	21.9	84	16.9	65	14.3	55
Σ Indispensable	184	199	108	142	77	124	67
% Indispensable	28%	52%		46%		42%	
Alanine		18.3		13.8		12.6	
Arginine		10.3		26.8		27.8	
Aspartic acid		40.9		34.3		35.6	
Glutamic acid		65.5		54.3		52.8	
Glycine		7.1		11.7		12.5	
Proline		21.7		13.3		13.8	
Serine		20.3		15.6		16.0	
Σ Dispensable	480	184	38	170	35	171	36
% Dispensable	72%	48%		54%		58%	
Total	664	383	58	312	47	295	44

^1^ Based on a mean total protein requirement of 0.66 g∙kg^−1^∙d^−1^. Adapted from FAO (2013).

**Table 3 nutrients-15-04211-t003:** The total (ΣAA) and indispensable (ΣIAA) amino acid peak concentration (C_max_) and area under curve (∆AUC_0-180_) response to the ingestion of 0.33 g∙kg^−1^ of protein isolates in healthy young adult men (*n* = 9).

	Cmax (µ·min·L^−1^)										
	Protein	Mean	SD	CI_95%_	*P*	η_p_^2^	Comparison		Diff	CI_95%_	*p*
ΣAA	WHEY	4232	279	429	<0.001	0.77	WHEY	PEA	272	645	0.104
	PEA	3960	292	449				FAVA	690	545	<0.001
	FAVA	3542	288	443			PEA	FAVA	418	537	0.005
ΣIAA	WHEY	2086	120	185	<0.001	0.96	WHEY	PEA	484	252	<0.001
	PEA	1602	146	225				FAVA	708	248	<0.001
	FAVA	1378	105	162			PEA	FAVA	224	206	<0.001
	AUC_0-180_ (mmol·min·L^−1^)										
	Protein	Mean	SD	CI_95%_	*P*	η_p_^2^	Comparison		Diff	CI_95%_	*p*
ΣAA	WHEY	177	19	29	0.014	0.87	WHEY	PEA	40	27	<0.001
	PEA	137	13	21				FAVA	66	52	<0.001
	FAVA	111	15	23			PEA	FAVA	26	32	0.003
ΣIAA	WHEY	125	11	18	0.023	0.95	WHEY	PEA	51	17	<0.001
	PEA	74	6	10				FAVA	72	35	<0.001
	FAVA	53	9	14			PEA	FAVA	21	24	0.002

## Data Availability

Data available upon request to corresponding author.

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
