# Peer review of "An Investigation of the Protein Quality and Temporal Pattern of Peripheral Blood Aminoacidemia following Ingestion of 0.33 g·kg−1 Body Mass Protein Isolates of Whey, Pea, and Fava Bean in Healthy, Young Adult Men"

_nutrients, 2023, doi:10.3390/nu15194211_

Round 1
Reviewer 1 Report
The manuscript “An investigation of the protein quality and temporal pattern of peripheral blood aminoacidemia following ingestion of 0.33 g⸱kg-1 body mass protein isolates of whey, pea, and fava bean in healthy, young adult men” is overall novel and with scientific significance. The study provide novel knowledge in comparison of protein quality between PSP and ASP. A rational way used in the manuscript could be useful in making advances in scientific and practical fields. However, the manuscript can be improved on some specific editing which are listed below:
1. Materials and Methods: Variety of the source of legumes is suggested to be supplemented in “2.2. Study Design”.
2. How are these proteins processed before ingestion? Are these proteins raw or thermal processed? Protein digestibility will be altered depend on the varied treating methods.
3. Repetition of the experiment data is missing in “statistical analyses” and is suggested to be supplemented.
4. There is no clear statement regarding Figure 3 (a)/(b) and (c)/(d). Please supplement the missing information at the corresponding legend.
Reviewer 2 Report
Major problem
The post prandial amino acid response to different dietary proteins is of interest but comparisons need to be made on a similar actual protein basis. The main finding of this paper is the lower post prandial response in terms of both total amino acids and indispensable amino acids for the pea and especially the fava compared with whey. However the differences in the responses in both the total and IAA shown in figure 1 are more marked than would be expected especially with fava raising the question of whether similar quantities of actual protein have been given. Table 2 shows that fava and pea contained less total amino acids (295 mg, 312mg) compared with whey (383 mg) consistent with the differences in their amino acid composition shown in supplementary table 1 in which the total amino acids as (g∙100-1 protein) amount to 109.76, 96.44 and 92.33 for whey, pea and fava. This implies that the actual protein content of the whey is overestimated and that of pea and fava isolates is underestimated. Unless this can be explained and rectified, possibly by correcting the post prandial responses to the same intake of total amino acids, then the results as shown do not represent true comparisons of these three protein isolates on an actual protein content basis. Although the compositional data appears to come from an authorative source, it doesnt make sense.
Minor issue
line 45 states As with most PSPs, legumes tend to be deficient in one or more essential amino acids, typically lysine and the sulphur containing amino acids methionine and cysteine. This is not true for lysine since its content in legumes is not limiting which is why legumes can complement cereals (which are lysine limiting).
Line 188. g/kg/d should be mg/kg/d
Round 2
Reviewer 2 Report
Your responses and text changes have gone someway to explaining the issue of how the use of a single N to protein conversion factor makes comparisons between dietary protein and amino acid intake problematic but you could do more and make your paper more useful.
1. You could include reference to FAO 2003 Food energy – methods of analysis and conversion factors: FAO FOOD AND NUTRITION PAPER 77 FAO Rome: as the most authoritative paper.
2 Your explanation of the differences in total amino acids per 0.33g protein in lines 174-178 could be expanded to include the actual %N for each of the three proteins which you can calculate on the basis of the weighted mean value for all the residues shown in table S1. For each amino acid the N content is the mwt-18 to account for the loss of 1 mol water in peptide bond formation. On this basis %N values for arg, gly, and tyr are 35, 25 and 8.6% so it will be interesting to see how the overall %N of the three proteins compare: FAVA and pea do contain more arginine and glycine than whey.
3. According to what your calculations show about the %N content you could then insert into lines 204 et seq "Matched for equivalent protein intake, (N*6.25),…………………….. and FAVA (+37%). However part of these differences reflected the lower total amino acid content per g nitrogen in Pea and especially FAVA compared with WHEY
